# Nonenveloped Avian Reoviruses Released with Small Extracellular Vesicles Are Highly Infectious

**DOI:** 10.3390/v15071610

**Published:** 2023-07-23

**Authors:** Zuopei Wang, Menghan He, Han He, Kyle Kilby, Roberto de Antueno, Elizabeth Castle, Nichole McMullen, Zhuoyu Qian, Tzviya Zeev-Ben-Mordehai, Roy Duncan, Chungen Pan

**Affiliations:** 1Laboratory of Molecular Virology and Immunology, Technology Innovation Center, Haid Research Institute, Guangdong Haid Group Co., Ltd., Panyu, Guangzhou 511400, China; 2Department of Microbiology & Immunology, Dalhousie University, Halifax, NS B3H 4R2, Canadakyle.kilby@ucalgary.ca (K.K.); arjde@dal.ca (R.d.A.); elizabeth.castle@ubc.ca (E.C.); nichole.mcmullen@dal.ca (N.M.); 3Department of Biochemical Engineering, University College London, London WC1E 6BT, UK; 4Bijvoet Centre for Biomolecular Research, Utrecht University, 3584 CG Utrecht, The Netherlands; z.zeev@uu.nl; 5Department of Biochemistry & Molecular Biology, Dalhousie University, Halifax, NS B3H 4R2, Canada

**Keywords:** avian reovirus, non-enveloped virus, extracellular vesicles, infectivity, Cryo-EM

## Abstract

Vesicle-encapsulated nonenveloped viruses are a recently recognized alternate form of nonenveloped viruses that can avoid immune detection and potentially increase systemic transmission. Avian orthoreoviruses (ARVs) are the leading cause of various disease conditions among birds and poultry. However, whether ARVs use cellular vesicle trafficking routes for egress and cell-to-cell transmission is still poorly understood. We demonstrated that fusogenic ARV-infected quail cells generated small (~100 nm diameter) extracellular vesicles (EVs) that contained electron-dense material when observed by transmission electron microscope. Cryo-EM tomography indicated that these vesicles did not contain ARV virions or core particles, but the EV fractions of OptiPrep gradients did contain a small percent of the ARV virions released from cells. Western blotting of detergent-treated EVs revealed that soluble virus proteins and the fusogenic p10 FAST protein were contained within the EVs. Notably, virus particles mixed with the EVs were up to 50 times more infectious than virions alone. These results suggest that EVs and perhaps fusogenic FAST-EVs could contribute to ARV virulence.

## 1. Introduction

Extracellular vesicles (EVs) are membranous vesicles of different sizes that are secreted by most cells and known as exosomes, microvesicles, or apoptotic bodies. In addition to being involved in regulating various physiological activities such as angiogenesis, cell proliferation, and gene expression [1,2,3,4], EVs also play an important role in intercellular virus transmission [5,6,7]. Traditionally, it has been thought that nonenveloped viruses release progeny viruses by directly destroying the cell lipid bilayer, but recent evidence suggests that they can be secreted outside the cells with the help of EVs. For example, the nonenveloped RNA virus, hepatitis A virus (HAV), rarely causes cytolytic infection, but it is released in the form of “quasi-enveloped” virions [8,9]; poliovirus hijacks autophagosomes to release clusters of virus particles packaged within phosphatidylserine-enriched EVs [10]; and hepatitis E virus (HEV) is released as quasi-enveloped virions via the multivesicular body pathway [11].

In recent years, researchers have found that several viruses from the order *Reovirales*, a large, diverse group of nonenveloped viruses with double-stranded RNA (dsRNA) genomes, are also released through EVs after infecting the host cells. For example, CaCo-2 cells infected with rhesus rotavirus can release different types of EVs that are associated with virus particles [12,13,14], while cultured human intestinal cells release rhesus rotavirus particles from the apical surface through a pathway that bypasses the Golgi and lysosomes [15]. Bluetongue virus (BTV), a member of the genus *Orbivirus*, can be released as either enveloped particles through plasma membrane budding or as non-enveloped virions by breaking the cell membrane [16]. As well, African horse sickness virus (AHSV), another member of the genus *Orbivirus*, appears to be exported by EVs [17]. Similarly, electron microscopy studies have shown that an insect reovirus that infects parasitoid wasps can be released by budding through the plasma membrane [18].

Avian reovirus (ARV) is a member of the order *Reovirales,* belonging to the *Orthoreovirus* genus in the family *Spinareoviridae*. ARV causes multiple diseases in poultry and leads to considerable economic losses for the poultry industry [19]. The ARV genome contains 10 dsRNA genome segments encoding eight structural proteins, λA, λB, λC, μA, μB, σA, σB, and σC, and four non-structural proteins, p10, μNS, σNS, and p17. Reovirus virions enter host cells through the endocytic pathway, during which the outer capsid protein σB is removed and the underlying μB protein is cleaved to generate intermediate subviral particles (ISVPs). The cleaved μB protein in ISVPs exposes membrane interaction motifs that permeabilize the endosome membrane to release the transcriptionally active virus core particle, which is surrounded by the inner capsid protein λA, into the cytoplasm where core particles transcribe viral mRNA to initiate viral gene expression [20]. ARV is a “fusogenic” orthoreovirus that induces syncytia formation using the p10 fusion-associated small transmembrane (FAST) protein to mediate cell–cell fusion [21], which contributes to the pathogenicity of the fusogenic reoviruses [22,23,24]. In this study, we set out to better understand whether EVs could affect the pathogenic mechanisms of these fusogenic non-enveloped viruses.

## 2. Materials and Methods

### 2.1. Cells and Vector

The quail fibroblast cell line (QM5) was maintained in medium 199 with Earle’s salts supplemented with 10% fetal bovine serum (FBS). Mouse fibroblast cell line L929 was cultured in Eagle’s Minimum Essential Medium with 10% FBS. The culture medium and FBS were obtained from Gibco^TM^, Thermo Fisher Scientific Inc. The ARV p10 FAST protein gene was cloned into the pcDNA3 vector as previously described [25,26], and the translational start site was optimized to the Kozak consensus sequence (GCACGATGG).

### 2.2. Viruses and Antibodies

ARV strain 176 was grown and titered in QM5 cells [27]. The virus titers were determined using a standard plaque assay [28]. All experimental infections were at a multiplicity of infection (MOI) of 0.2. The rabbit polyclonal antisera against FAST protein p10 or ARV outer capsid proteins σB and µB, and chicken antisera that recognize all virus structural proteins obtained from ARV-infected animals, were developed by the laboratory of Roy Duncan at Dalhousie University [25]. Monoclonal antibodies against TSG101 (Cat. No. sc-7964) and Hsp70 (Cat. No. sc-24) were obtained from Santa Cruz Biotechnology, Inc. All of the secondary antibodies were obtained from Thermo Fisher Scientific Inc., Ottawa, Canada.

### 2.3. EV Isolation

EV isolation was performed with a similar method as previously described [29,30]. Briefly, six T175 flasks of QM5 cells infected with ARV at MOI of 0.2 were cultured in the vesicle-collection medium. The vesicle-collection medium was DMEM-supplemented with pre-cleared 10% FBS, which was prepared by dilution with DMEM to 50% FBS followed by centrifugation at 100,000× *g* for 16 to 18 h [31]. For isolation of EVs, the culture supernatant from virus-infected cells was collected at the indicated time, and the cell debris was removed by successive centrifugations at 2000× *g* for 10 min at 4 °C and at 10,000× *g* for 30 min at 4 °C. The supernatant from the last spin was transferred to Ultra Clear tubes (Beckman Coulter Life Sciences) to isolate the EVs by centrifugation at 100,000× *g* in a SW41 Ti rotor for 120 min at 4 °C. The vesicle-enriched pellet (named as p100) was resuspended in 500 µL Phosphate Buffered Saline (PBS) and loaded on the top of an 8–44% iodixanol step gradient (OptiPrep™ Density Gradient, Sigma, Cat no. D1556) in an ultra-clear SW60 ultracentrifuge tube (Beckman Coulter Life Sciences). The gradients were centrifuged for 36 h at 105,000× *g* at 4 °C using slow acceleration and no brake deceleration. The fractions were harvested from the top of the gradients (usually 21 fractions). The densities of each fraction were measured with a refractometer. The 21 fractions were run in the three separate gels at the same time with two electrophoresis tanks, and the same antibody-probed membranes were kept side by side in the imager for capturing the blots from different membranes into one image. To avoid the overlapping fractions of EVs and virions, the fractions 9 to 13 were collected and gently mixed well as EV samples, and the fractions 19 to 21 was also mixed well and used as virions samples; fractions 14 to 18 were skipped.

### 2.4. Western Blotting

Protein samples were run into Tris-glycine 12.5% polyacrylamide gels, transferred to a PVDF membrane, and probed with the indicated primary and secondary antibodies. Blots were developed with an ECL plus detection reagent (GE Healthcare). Images were captured on an Amersham Typhoon 9400 Variable Mode Imager (GE Healthcare). The intensity of protein bands was quantified using ImageJ1.49e software. Only lanes which showed a decline proportionate to the two-fold dilution series were used for calculations. To compare the ratio of μB or σB to λA between the fractions of EVs and pure virus particles, the ratio of each protein to λA in the virion fraction was given a value of 1 and EV fraction ratios were calculated in relation. The dilutions of the primary antibodies in immunoblots were 1:5000 for chicken-anti-ARV, and rabbit-anti-σB or µB, 1:2500 for rabbit-anti-p10, and 1:500 for mouse-anti-TSG101 or Hsp70. The dilutions of all secondary antibodies were 1:5000.

### 2.5. Sandwich ELISA

The p100 pellet was prepared with 6 T175 flasks of ARV-infected QM5 cells as indicated in Section 2.3 and resuspended in 500 µL pre-cooled PBS. The 250 µL suspension was added with less than 5% volume (about 10 µL) of PBS-pre-diluted Triton-100, with a final concentration of 1% Triton-100, and incubated for 15 min at room temperature, while the control sample was prepared by using the rest of the 250 µL suspension and adding PBS in the same volume as the Triton-100 solution. At the end of treatment, only the Triton sample was sonicated. For the ELISA, a 50 µL sample was added into each well of 96-well high-binding polystyrene plates (Costar; Corning, Inc., Corning, NY, USA), which were pre-coated with rabbit antiserum against ARV σB protein. The plates were incubated at 37 °C for 45 min and washed with PBST (PBS plus 0.5% Tween 20) four times. Subsequently, the plates were incubated with chicken antiserum against ARV at 37 °C for 45 min, washed four times, and incubated with horseradish peroxidase (HRP)-labeled goat IgG against chicken IgY (Novex, Frederick, MD, USA) at 37 °C for 45 min. After the same washing step as above, the substrate, 3,3,5,5-tetramethylbenzidine (TMB Substrate Kit; Thermo Fisher Scientific, Rockford, IL, USA), was added and incubated in the dark for 10 min. The plates were measured at 450 nm using an ASYS Expert 96 plate-reader (Montreal Biotech Inc., Kirkland, QC, USA). The antiserums and HRP-labeled IgG were used at a dilution of 1:2000.

### 2.6. Transmission Electron Microscopy

Cell culture supernatants were collected 24 h post-ARV infection, as described in Section 2.3. The pre-cleared supernatant was further centrifuged at 100,000× *g* for 70 min to pellet EVs. The pellets were re-suspended in 500 µL PBS and centrifuged at 100,000× *g* in a fixed angle rotor (Beckman Coulter Life Sciences) for 35 min to obtain a pellet with EVs, which were dislodged from the centrifuge tubes and embedded in 20 µL of 3% Ultralow SeaPrep agarose dissolved in an exosome-free complete cell culture medium. Agarose-embedded EVs were then fixed for 2 h at room temperature in phosphate-buffered 2.5% glutaraldehyde containing 1% acrolein and 0.25 M sucrose. After fixation, EVs were incubated with 2% OsO_4_ for 2 h, followed by overnight en bloc staining in 0.25% uranyl acetate (positive stain). After dehydration using acetone, samples were embedded in Epon-Araldite. Thin sections were examined with a JOEL JEM 1230 transmission electron microscope operating at 80 KV. Virion fractions were also concentrated with the same method as used for EVs but followed by adsorbing the virions onto carbon-coated Formvar grids and negative staining with uranyl acetate as previously reported [32].

### 2.7. Quantitative Polymerase Chain Reaction (qRT-PCR) Assay

To normalize the virus gene concentration in the fractions which were collected from the gradients after 36 h centrifugation (as described in Section 2.3 EVs isolation), 5 × 10^4^ cells (L929) per well were plated in a 12-well plate, which were directly lysed without any treatment after 16 h with 200 µL lysis buffer. The total RNA was extracted from the mixture of 50 µL of the OptiPrep fractions and 200 µL of L929 cell lysate by using the RNeasy^®^ Plus Micro Kit (QIAGEN). The mixing of the L929 cell lysate with the EV fraction increased the ARV genome yield with the micro-column extraction of the cell lysates obtained at 4 and 18 h post-treatment with the standardized inocula, and it also provided L929 RNA as the internal control gene. The reverse transcription was performed by using forward primer F634 (CAAATGCAAACGACCACCAC) for ARV S3 negative strand RNA (only virus genome RNA was amplified) and Oligo(dT)18 primer (Thermo Fisher Scientific Inc.) for cellular mRNA. A quantitative PCR was carried out with reaction ingredients containing the reverse transcripts, SYBR^®^ Green qPCR Supermixes (Bio-Rad Laboratories Co., Ltd., Hercules, CA, USA), and the primer pairs with the forward primer F735 (TGTAAAGCTTGCGAATGCTGAC) and reverse primer R976 (GCTCCATTCCTGTAGCGCAT) for the ARV S3 gene and the forward primer TGCACCACCAACTGCTTAGC and reverse primer GGCATGGACTGTGGTCATGA for GAPDH as an internal control.

The virus gene in the infected QM5 cells were also quantified with a similar method, but the forward primer TGGAGAAAATCTGGCACCACACC and the reverse primer GATGGGCACAGTGTGGGTGACCC for actin were used as an internal control instead of GAPDH.

The PCR reactions for amplification of the S3 gene, GAPDH, or actin were performed using the same parameters, where the annealing temperature was 59 °C and the running cycles were 40.

### 2.8. Infectivity Assay

The relative infectivity of the EV and the virion fractions were measured by qRT-PCR assay at early and mid times post-infection using inocula standardized by the concentration of the μB outer capsid protein and independently confirmed by plaque assay using inocula standardized based on ARV genome equivalents.

To measure the infectivity of the fractions, the inoculum was standardized to equivalent concentrations based on the outer capsid protein μB as quantified by Western blotting with the serial diluted fractions. The QM5 cells in 12-well plates were infected with an equal volume of the opti-DMEM supplied with fractions containing either EVs or pure virions. The volume of virion fraction used was 2 µL, and the volume of EV fraction ranged from 50–250 µL based on the ratio of μB of the EV fraction and of the virion fraction calculated by Western blotting. The plates were washed twice with PBS after a 30 min incubation at 37 °C. After this, the virus negative RNA in the cells were quantified by qRT-PCR at early (4 h) or mid (18 h) times post-infection. Protein quantification and infectivity experiments were independently repeated 4 times, and the means and standard deviations at each time point were calculated.

For the plaque assay to measure the relative infectivity of the EV and virion fractions, the inoculum of the EVs or pure virions from OptiPrep gradient (collected as in Section 2.3 EVs isolation) were normalized based on the virus negative genome strand for S3 gene by qRT-PCR (as described in Section 2.7). Generally, 2 µL of the virion fraction was equal to 50 to 250 µL of the EV fraction. Then the QM5 cells in 12-well plates were infected with the serial diluted fractions containing either EVs or pure virus particles with an equal amount of virus genome RNA, followed by adding the soft agar and incubating for 96 h in an inverted position. Then the agar was gently removed, and the cells were stained with crystal violet to count the plaques [28].

### 2.9. Mass Spectrometry

The EV fractions were run on 15% polyacrylamide gels, stained with Gelcode^TM^ Blue Stain Reagent (Thermo Fisher Scientific Inc.) and washed with distilled water 5 times for 5 min each, until the protein bands were clearly observed. The protein bands of interest were excised and analyzed by mass spectrometry.

### 2.10. Cryo-Electron Microscopy

A 3 μL aliquot of the EV fractions was pipetted onto a glow-discharged holey carbon-coated copper electron microscopy grid (C-flat, Protochips). The drop was blotted, and the sample was vitrified by plunging it into liquid ethane (−183 °C). Projection images were recorded on a Tecnai F30 Polara TEM (FEI) operated at 300 kV and equipped with a GIF2002 post column energy filter (Gatan) operated in zero loss mode at defocus settings between −4 μm to −6 μm [33].

## 3. Results

### 3.1. ARV-Infected Cells Release Viral Proteins in EVs and the EV Fractions Are Infectious

Our initial goal was to determine whether the ARV p10 integral membrane protein was released in exosomes. To do so, we infected quail (QM5) cells with ARV strain 176 and used differential and buoyant density centrifugation to isolate small EVs (Figure 1A). The results displayed that QM5 cells infected with ARV started to fuse to each other at 16 h post-infection, and the syncytia gradually expanded over time (Figure 1B). At 40 h, most cells in the culture dish were fused together. The small EVs were collected from the supernatant of the virus-infected cells at various time points post-infection (16, 20, 24, 28, and 40 h) by differential centrifugation with a final pelleting at 100,000× *g* to obtain the p100 fraction, as described in the Section 2. The p100 from 24 h post-infected cells was analyzed by immunoblotting following treatment with Triton X-100, sonication to break the EV membrane, and re-pelleting at 100,000× *g* [12]. Immunoblotting data showed that most of the transmembrane protein p10 was released into the supernatant (Figure 1C), indicating that the EV membrane and p10 were solubilized under the treatment, and in turn indicating that the ARV FAST protein was released in the EVs. Immunoblotting using the chicken antiserum specific to the three major structural proteins, λA, µB, and σB, revealed that most of these major capsid proteins were present in the post-sonicated pellet, implying that they were present in virions (Figure 1C). This indicated that the virions and EVs were both present in the p100 fraction. Furthermore, when ELISA was used to detect the presence of outer capsid protein σB (representing the full virus particle) in the p100 fraction obtained from ARV-infected cells at 24 h and 28 h post-infection, and with or without the treatment with 1% Triton X-100, significantly more outer capsid protein σB was observed when the EV membrane was broken with Triton X-100 (Figure 1D). This suggested that σB might be present within the EVs.

The p100 pellet was further fractionated with OptiPrep buoyant density gradients and analyzed by Western blotting. The EV fractions were identified by immunoblotting using the EV markers Tsg101 and Hsp70, and these fractions also contained the ARV p10 FAST protein (Figure 1E). These results implied that this non-structural viral protein is released as an integral membrane protein in EVs. As expected, we observed the presence of the major ARV outer capsid protein σB in the densest lower Optiprep fractions, corresponding to a density of 1.2462 gm/mL where the intact virions pelleted (ARV virions have a density of ~1.36 g/cm^3^) [34]. However, σB was also detected in large quantities in the less dense upper fractions extending into the EV fractions (Figure 2D). Interestingly, when EV fractions 9–13 were collected at 24 h post-infection and pooled and then added to non-infected QM5 cells, extensive syncytia formation occurred, similar to what was observed when cells were treated with the high-density ARV virion fractions from the OptiPrep gradients (Figure 1F). These results indicated the EV fractions contain infectious material.

### 3.2. EV Fraction Is a Mixture of EVs and Virions and Is Highly Infectious

To examine the basis for the infectious nature of the EVs, electron microscope techniques were applied to observe the vesicles in the EVs. The transmission electron microscope images displayed that the electron-dense centers in the EVs were about the size of reovirus virions (Figure 2A top) and appeared to contain angular structures reminiscent of reovirus virions (2 rows of images in Figure 2A bottom). However, EVs from uninfected cells showed similar electron dense centers. To conclusively determine whether the infectious nature of the EV fraction reflected the presence of vesicle-encapsulated virions, EV fractions were analyzed by cryoEM tomography. The majority of the ~100 nm EVs contained irregularly shaped inclusions, but these inclusions were clearly not reovirus virions (Figure 2B). The cryoEM tomogram (Appendix A) also indicated that the free virus particles were not encapsulated within the vesicles.

To further characterize the nature of EVs and how they might contribute to ARV infectivity, the relative infectivity of the OptiPrep fractions containing either EVs (fractions 9–14) or virions alone (fractions 19–21) was determined by their application onto uninfected QM5 cells. To standardize the virus inocula, the EVs and the virion fractions were normalized based on the major outer capsid protein μB, which was quantified by serial dilutions to obtain a linear detection range by Western blotting (Figure 2C). QM5 cells were treated with standardized inocula and cells were harvested at early (4 h) and mid (18 h) times post application of EVs or virion fraction. The relative infectivity of the two treatments was quantified by qRT-PCR using primers specific for the minus strand of the S3 dsRNA genome segment, indicative of the amount of progeny virus genome replication. In four independent experiments, at 4 h post-infection, which is a limited time for viral genome replication to have occurred, there was only a modest increase in the infectivity of the EV fraction relative to the virion fraction (1.6 ± 0.3 fold) (Figure 2D). However, by 18 h post-infection, when progeny viral genome replication is well underway but before secondary infections can be established, the fold infectivity of EV fractions over the virion fraction ranged from 9.7–47 fold (26.9 ± 18.9 fold) (Figure 2D).

Free viral proteins or genes may exist in EVs, in addition to the extra-vesicular virions present in these fractions; therefore, testing only the viral proteins to calibrate the content of virus inoculation may skew the results. We therefore corroborated the results obtained using μB to standardize the inocula by using qRT-PCR to standardize genome equivalents in the EV and virion inocula. In these experiments, we also directly assessed the relative infectivity of the EV and virion fractions by plaque assay. The results exhibited that EVs were significantly more infectious than the virion fraction (Figure 2D). The infectivity of the EV fractions ranged from a 33–50.8 fold (40.7 ± 9.9 fold) increase relative to the virion fraction in four independent experiments (Figure 2D) when the amount of virions was normalized based on the virus genome and the plaque assay was used to quantify the infectivity.

### 3.3. Viral Proteins and Full Virus Particles Exist in EV Fractions

While analyzing the virion and EV fractions by Western blotting, we noted a difference in the relative ratios of the major capsid proteins. The chicken antiserum was obtained from virus-infected birds and recognizes the major core (λA) and outer capsid (σB, μB) structural proteins, including the minor σC spike protein [27]. Quantification of the band intensities of the σB and λA proteins using serial dilutions of the virion and EV fractions from three independent experiments revealed that the EV fractions contained less of the σB outer capsid protein relative to the λA core protein (Figure 3B). This was also apparent visually for the μB outer capsid protein. The lower outer capsid:core protein ratio is the expected protein profile of orthoreovirus core particles, which lack all three outer capsid proteins (Figure 3A top and middle).

We also noted an increase in the detectible levels of a protein present in EVs that reacted with the chicken anti-ARV serum and which migrated slightly faster than the σB protein (Figure 3B), the location where the σC spike protein migrates [27]. This region in the SDS-PAGE gel was excised for mass spectrometry and the results showed that the band contained the σC protein (Appendix A). The σC protein is present as a trimer at the vertices of the icosahedral capsid, although not all vertices may have an σC trimer [35,36], which means that there are only 36 copies or less of σC present in ARV virions. Under the protein loads used for Western blotting, this protein was undetectable in the virion fraction, and while the levels were variable between experiments, the σC band was always detectible in the EV fraction and was frequently a prominent band (Figure 3B).

The protein profiles were also analyzed following the Triton X-100 solubilization and re-centrifugation of the EV and virion fractions. As shown (Figure 3D,E), the detergent treatment resulted in an equivalent σB outer capsid: λA core protein ratios. These results suggested that a soluble λA protein may be present in the EV fraction (i.e., the removal of the soluble λA protein by the detergent left all the remaining virus proteins in the virions but not in the core particles). In contrast, the excess σC spike protein present in the EV fraction was largely resistant to detergent extraction (Figure 3D). The same phenomenon was also observed in Triton-treated p100 samples (Appendix A). We conclude that, in addition to the p10 FAST protein likely embedded in EV membranes, EVs contain a soluble λA protein and an insoluble form of the σC spike protein. Whether any of these viral proteins contribute to the infectivity-enhancing properties of EVs released from ARV-infected cells remains to be determined.

## 4. Discussion

We demonstrate that ARV-infected quail cells release abundant virions, some of which co-sediment in OptiPrep gradient fractions that contain small (~100 nm) diameter EVs. CryoEM tomography and Western blotting with and without detergent treatments conclusively demonstrated that these EVs did not encapsulate ARV virions or cores, but they did contain the p10 integral membrane protein and appeared to encapsulate two viral structural proteins, the major core protein and the viral cell attachment protein. Most noteworthy was our observation that the virions plus EVs are possibly as much as 50 times more infectious than the virions alone. These observations may impact our understanding of ARV virulence and pathogenesis.

In this study, the virions appeared in the EV fraction probably as coincidental overlap rather than because the virions are somehow unique in their buoyant density or because they specifically interact with EVs, although further studies are required to formally exclude the latter possibilities. The σB protein displayed a very broad distribution in the OptiPrep gradient that peaked after the virion fraction and tailed off into the EV fractions (Figure 1E), and there was a relatively small proportion of virions present in the EV fraction relative to the virion pellet. The virus could be released from lysed cells or could be released in larger EVs as with mammalian orthoreovirus (MRV) [37]. These larger microvesicles would have been mostly excluded from our analysis by the 10 K centrifugation step. If these larger microvesicles were ruptured during purification, it would release the encapsulated virions and would co-pellet with the small EVs. Therefore, it seems most likely that the virions in the EV fraction were due to limitations in the gradient resolution. Nonetheless, this coincidental finding was what led us to observe the effect of EVs on enhancing the infectivity of nonenveloped virions.

We thought that the increased infectivity might reflect core particles encapsulated inside the EVs, based on the reduced outer capsid:core protein ratio. Cores themselves are not infectious since they lack the membrane-penetrating μB cleavage products. However, if they were present inside the EVs, then perhaps EV-to-cell fusion could directly deliver core particles into the cytoplasm where they would be infectious and contribute to the increased infectivity imparted by the EVs. The presence of the p10 FAST protein in Evs could promote EV-to-cell fusion and core delivery, leading to enhanced infectivity. However, this would probably require the core to be inside the EV and cryoEM tomography did not detect ARV cores. Detergent extraction of Evs also restored the outer capsid:core protein ratio, suggesting the presence of viral particles, not cores. Furthermore, the infectivity assays based on genome equivalents would have accounted for core particles.

Our results suggest that, in addition to EV-embedded p10 FAST protein, EVs also contain the λA and σC proteins. The λA protein was released by detergent extraction, suggesting that it is encapsulated in a soluble form within EVs, while σC still pelleted after detergent extraction, suggesting that it remained in an insoluble form. What, if anything, these viral proteins could be doing inside EVs and how they ended up in Evs is not clear. During ARV replication, λA is the first structural protein that is recruited into the virus factory by μNS for the virus core shell formation [38]. It is unclear how this property could be related to the enhanced infectivity phenotype of Evs from ARV-infected cells. The σC protein is a soluble trimer whose amino-terminus interacts with residues present in the λB turret protein present at the 12 vertices of the icosahedral virion, from where it can bind to receptors on the cell membrane and mediate virus entry [39]. It is conceivable that excess σC might aggregate but still be capable of binding to receptors and interacting with EVs to promote EV attachment to cells, although σC is a highly soluble protein and it is unclear what might cause it to aggregate. Additional studies are needed to test this speculation and the effects, if any, of σC and/or λA on the infectivity-enhancing properties of the EVs released from ARV-infected cells.

Even in the absence of ARV cores in EVs, the presence of p10 in EVs provides some interesting possible means of promoting virion infectivity. The p14 FAST protein has been shown to promote liposome-to-cell fusion [40,41], suggesting that the p10 FAST protein could promote EV-to-cell fusion. Fusogenic EVs could increase infectivity by releasing the contents beneficial for infection from the EVs into the target cells, such as viral mRNA [42] or something that promotes the infection of virions by blocking innate immunity factors [43]. The FAST proteins and syncytia formation have long been known to be virulence factors of the fusogenic orthoreoviruses [22,23,27], and they promote oncolytic virotherapy [44]. These phenotypes are thought to be linked to the only described biological function of the FAST proteins—syncytia formation. Our new results suggest that the fusogenic EVs may need to be considered when thinking about FAST proteins and pathogenicity or oncolytic virotherapy.

Several studies indicate that viral proteins from non-enveloped viruses in the EVs may assist virus-EV complex formation, virus releasing, or virus infection. For example, HAV-infected cells released the extracellular vesicles, called exosome-like vesicle-coated HAV (eHAV), contained in the capsid surface structural protein pX. The C terminal of pX is involved in eHAV release through interaction with the ALIX (apoptosis-linked gene 2-interacting protein X) protein [45]. Capsid protein VP2 of another non-enveloped virus, the Bluetongue virus (BTV), interacts with the viral non-structural proteins NS3 and NS3A and acts as a component of the cellular ESCRT (exosome sorting complexes required for transport) pathway to mediate the release of BTV in budding form [17]. Furthermore, the ORF3 protein of HEV (hepatitis E virus) is responsible for the membrane-associated HEV particles released through the interaction between its Pro-Ser-Ala-Pro (PSAP) motif and Tsg101 [46]. EVs carrying different biomolecules such as proteins, carbohydrates, lipids, and nucleic acids may enhance the infectivity of the viruses by regulating immunity, manipulating the microenvironment, enhancing replication, and mediating the transfer of the viral genome [42,47,48,49]. The particular biomolecules that may promote infectivity of the ARV-EV complex remain to be investigated.

## Figures and Tables

**Figure 1 viruses-15-01610-f001:**
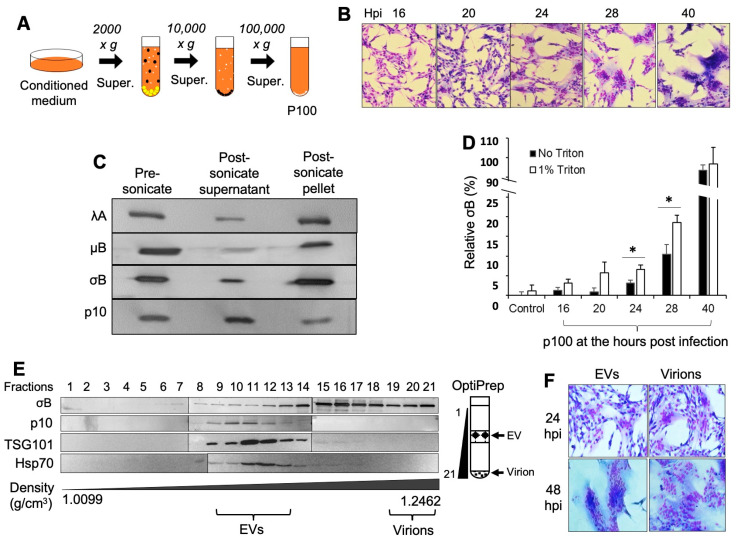
ARV-infected cells release virus proteins in EVs and the EV fractions are infectious. (**A**) The conditioned culture medium was collected by differential centrifugation to obtain the p100 pellet. (**B**) The QM5 cells were infected with ARV and Giemsa-stained at the indicated time points. The culture medium was collected at the same time points to harvest p100 as per the method indicated in Section 2.3. (**C**) The 24 h p100 sample was treated with 1% Triton and sonication, then re-pelleted by 100,000× *g* (the method in Section 2.5). ARV proteins in the supernatant and pellets were analyzed by immunoblotting with the chicken antibody against ARV virions (only λA blots displayed) and rabbit antibodies against μB, σB, or FAST protein p10. Representative images from one of two experiments are shown. (**D**) The p100s were treated with or without 1% Triton and the outer capsid protein σB was measured by sandwich ELISA. The OD450 absorbance data for the 40 h sample with Triton were set as 100%, and the control sample without Triton was set as 0% to calculate the relative amount of outer capsid protein σB for the rest of the samples. The conditioned culture mediums collected from the uninfected cells were treated with the same methods and used as the control samples. Results are the mean and standard deviation from *n* = 2 independent experiments. Student’s *t*-test was used for statistical analysis, * represents *p* < 0.05. (**E**) The p100 fraction collected at 24-h post-infection was further fractionated on OptiPrep gradients to separate denser virus particles from EVs, and gradient fractions (1–21, top-bottom, as shown in right cartoon) were analyzed on Western blots probed with antibodies against the major outer capsid protein (σB), FAST protein (p10), or exosome markers (TSG101, Hsp70). Fractions corresponding to EVs and virus particles are labeled in the image. (**F**) Quail cells were treated with equal volumes of the EVs (fractions 9–14) or virions (fractions 19–21) from OptiPrep performed on p100 obtained from conditioned media of 24 h post-infected cells. Cells were Giemsa-stained at 24 h or 48 h post treatment and imaged by light microscopy at 200× magnification. Representative images from one of two experiments are shown in panel 1 (**F**).

**Figure 2 viruses-15-01610-f002:**
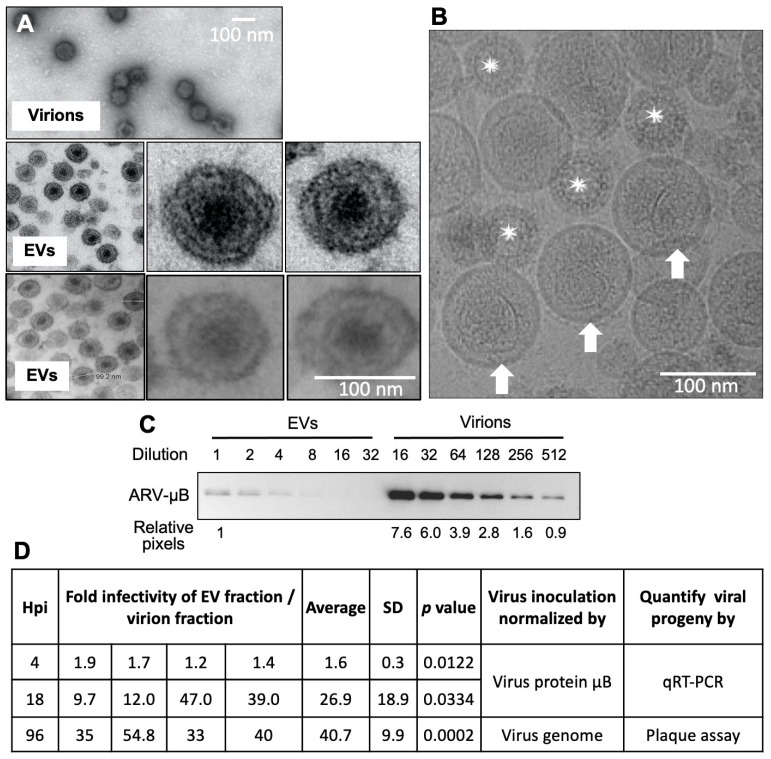
EV fractions containing virions are highly infectious. (**A**) Electron micrographs of the sectioned and negatively stained virion fraction (top) or positively stained EV fractions (bottom two rows; each row of the images was obtained from independent experiments). The right insets in each row show a higher magnification of the single EVs from the left image. (**B**) Cryo-EM projection images of the EV fraction. Stars indicate mature virus particles, and the arrows point to the EVs that have an irregular density in them instead of a virus particle. Scale bars, 100 nm. (**C**) Serial dilutions of the EVs and virion fractions were run on SDS-PAGE and the outer capsid protein μB was quantified by Western blot to standardize equivalent concentrations for infectivity assay. For this experiment, a ~500-fold dilution of the virion fraction gave equivalent concentrations of μB as in the EV fraction. (**D**) The EVs and virion fractions containing the same amount of virus protein μB as determined by Western blotting or genome equivalents as determined by qRT-PCR of the negative genome strand were used to infect QM5 cells. Virus gene transcripts were quantified by qRT-PCR at 4 or 18 h post-infection as a measure of relative infectivity for cells treated with fractions standardized based on the μB concentrations. Experiments using standardized genome equivalents for inocula were serially diluted and directly plaqued on QM5 cells. Plaques were counted at 96 h post-infection to calculate the fold infectivity. Results are shown for four independent experiments and the mean ± SD for these experiments. Statistical significance was determined by Student‘s *t*-test.

**Figure 3 viruses-15-01610-f003:**
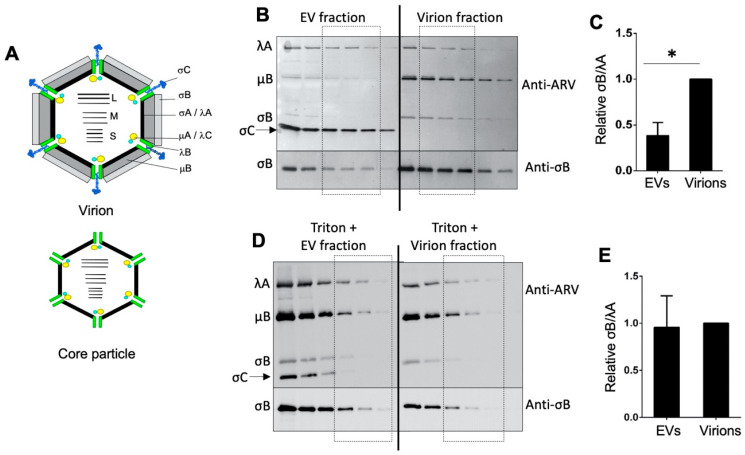
Soluble virus proteins and full virus particles are included in EV fractions. (**A**) The schematic diagrams for ARV virion composed of the core proteins, λA, λB, λC, µA, and σA surrounding the ten dsRNA genome segments (large (L), medium (M), and small (S)), and the major outer capsid proteins, µB, σB, σC. Note: µB refers to the cleaved mature form of the precursor µB protein. The core particle indicates the major structural proteins detected in the accompanying Western blots. (**B**) Two-fold dilutions of equal volumes (20 µL) of EVs and virion fractions were analyzed by Western blotting using rabbit polyclonal antiserum specific to the major outer capsid protein σB or using polyclonal antiserum obtained from ARV-infected chickens that recognizes all of the virus proteins including the major λA, μB, and σB capsid proteins and the σC spike protein. The pixel intensities in the boxed region of the Western blots were normalized to the λA core protein to quantify the relative levels of the σB outer capsid protein. (**C**) The bar graph was generated by analysis of the Western blots from panel B using data obtained from three independent experiments. Student’s *t*-test was used for statistical analysis; * represents *p* < 0.05. (**D**,**E**) A similar experiment and analysis as in panels (**B**,**C**), except the Western blot, was performed after the treatment of both the EVs and virion fractions with 1% Triton to break the membrane of the EVs followed by re-centrifugation to remove solubilized proteins.

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
