# Peer review of "Nonenveloped Avian Reoviruses Released with Small Extracellular Vesicles Are Highly Infectious"

_viruses, 2023, doi:10.3390/v15071610_

Round 1
Reviewer 1 Report
Many viruses, including some viruses in the order Reovirales, are released from cells in association with extracellular vesicles (ECVs). Avian reoviruses (ARV) are non-enveloped, fusogenic viruses belonging to the order Reovirales that can be pathogenic in birds, including poultry. Wang et al. used electron microscopy and immunoblotting of “ECV" and “virus” fractions released from infected cells, enriched by differential centrifugation and gradient ultracentrifugation, to understand the nature of ARV exit from quail cells (CM5). They report that following ARV infection, CM5 cells release small (~100 nm), electron dense ECVs with which both free ARV proteins and intact ARV particles associate. To measure the infectivity of “ECV” and “virus” fractions, the authors used inocula that were normalized based on outer capsid protein muB or on genome copies to infect cells and quantified viral transcription using RTqPCR or viral progeny by plaque assay. They suggest that association with ECVs enhances ARV infectivity up to fifty-fold. Based on the ratios and identities of ARV structural proteins detected in “ECV” and “virus” gradient fractions in the presence or absence of detergent treatment and on cryo-EM images of “ECV” fractions, they conclude that free ARV proteins are present within ECVs, but mature ARV particles associate with but are external to ECVs.
Broad comments:
Viral egress mechanisms are understudied, particularly for non-enveloped viruses. Virus egress in association with ECVs and its impacts on infection are of broad interest, as they can inform our understanding of virus biology and pathogenesis as well as vaccine design. In the current study, Wang et al. made interesting observations regarding ARV egress, and some conclusions are supported. For example, experiments to test whether the presence of ECVs enhances ARV infectivity on a per genome basis appear to have been conducted in multiple independent experiments and indicate that the presence of ECVs enhances ARV infectivity, measured by the output of infectious ARV by plaque assay. Although the mechanism is unclear, this is an exciting finding. However, in some cases missing experimental details or controls, limited rigor, or other factors call the authors’ conclusions or their significance into question and could be improved. For example, while ARV-infected CM5 cells released electron-dense small ECVs, the nature of ECVs released by mock-infected CM5 cells was not reported for comparison. Viral protein ratios used to make conclusions about protein incorporation into ECVs were quantified based on immunoblotting experiments that were conducted only once, rather than in multiple independent experiments. Although the authors describe the relationship between mature ARV particles and ECVs as an “association,” additional experimentation is needed to convincingly indicate a specific interaction, rather than a coincidental overlap in density. Comments below detail major and minor concerns.
Specific comments:
Introduction: Rotavirus, reoviruses, and orbiviruses are considered members of the order Reovirales rather than the Reoviridae family, a recent change in virus classification.
The authors might consider using the abbreviation “EV” rather than “ECV” throughout the manuscript, as it is widely used in the field, including by the International Society for Extracellular Vesicles (ISEV).
The authors should consider including Santiana et al. Cell Host Microbe, 2018 as a reference for rotavirus egress in association with EVs.
Methods: Please briefly mention whether FBS was diluted prior to pre-clearing and potential problems arising due to the use of FBS in culture during ECV collection (PMCID: PMC7830136).
Fig. 1B: The inclusion of TEM images of ECVs in tetherin-overexpressing, mock-infected cells is an important control. Are these ECVs electron dense or electron lucent? If electron lucent, the significance of the description of ECVs generated by infected cells will be clearer.
Page 4, “The ECVs (p100) were collected as indicted in Figure 2B at various time points post-infection” should read “…as indicated in Figure 1D...”
Fig. 1D: Is there a reason the authors decided not to investigate ARV virus or protein association with other fractions, including the 10,000 x g fraction? Multiple viruses are released in association with secretory autophagosomes, and rotavirus is released in association with microvesicles. These EVs may be found in the 10,000 x g pellet.
Fig. 1E: Which statistical analysis was applied? What P values do the asterisks represent? Evidence of ECV disruption by TX-100 treatment, perhaps by loss of ECV marker signal by immunoblotting after re-pelleting following treatment or by EM imaging after treatment, could strengthen conclusions regarding these and following results.
Fig. 1F: SigmaB protein is detected in protein fractions across much of the gradient, with an apparent peak around fractions #14-17. The presence of ECVs in fractions #8-13 does not correspond with an increased concentration of sigmaB protein. Thus, the ECVs and ARV particles may simply have overlapping densities. Although it would be ideal to resolve “ECV” and “virus” fractions on the same immunoblot to ensure equivalent transfer and exposure, the raw data suggest that the authors exposed each set of immunoblots for a given marker simultaneously. This methodology should be clarified in the main manuscript. Stronger support could be provided by a quantified comparison of the protein bands from at least three independent gradient fractionation and immunoblotting experiments.
Figs. 2B-C: It is unclear why a different normalization strategy, virus genomes by RTqPCR, and a different output determinant, virus titer by plaque assay, were used for the 96-h time point than at the 4 h and 18 h time points. If different inputs and outputs were used, it is not appropriate to directly compare the results of these assays in Fig. 2C. However, for those experiments that were conducted with identical inputs and outputs, please apply statistical analyses. Based on the text, it appears that the authors applied an ~ 500 times greater volume of “ECV” fractions than “virus” fractions to cells prior to quantifying virus transcripts. Please clarify the ratios of “ECV” to “virus” fraction volume used for “virus genome” inocula. Please clarify whether the four repeated experiments reported are technical replicates using “ECV” and “virus” fractions from the same gradient or independent experiments from four independently prepared gradients.
Fig. 3A. Some of the raw data for this figure appears to be missing.
Fig. 3A-B and 3D-E: ECV preparations can be variable. The “n = 3” in the legend seems to refer to analysis of the three lanes of each single immunoblot that is shown. If this is true, immunoblots should be repeated and quantified for at least three independent experiments for 3B and 3E to ensure rigor when making conclusions about the ratios of proteins present in different fractions.
Fig. 4: Imaging the “virus” fraction by at least one method would be highly informative. It would also be nice to see more than one image of each preparation.
By imaging diluted “ECV” gradient preparations using cryo-EM, the authors may gain insight into whether ARV particles are tethered to ECVs or simply present alongside them. More definitive results could be provided by immunoprecipitation studies to determine whether ARV particles pull down ECVs, checking for ECV markers by immunoblot, and whether ECVs pull down ARV particles, perhaps using TIM4- or annexin V-conjugated beads, which bind phosphatidylserine.
Discussion paragraph 4: Connections between p10/cell fusion and NFkB with results presented in the current study are somewhat unclear.
The discussion section would benefit from discussion of how syncytia formation and ECV-assisted infection might coincide. If the primary mode of ARV cell-cell transmission is through cell-cell fusion, then under what conditions might EV-assisted infection be important?
Discussion paragraph 5: It would be helpful if the authors would explain why sigmaC was not detected in the virus fraction and propose a potential mechanism(s) through which the presence of sigmaC in ECVs might enhance ARV infection.
Discussion paragraph 6: The authors hypothesize that free virus structural proteins present in ECVs may promote internalization of viral particles into cells. However, data presented throughout the manuscript suggest that some free viral structural proteins are enclosed within ECVs (Fig. 3), and some of the viral structural proteins detected in the ECV fraction are incorporated in mature virus particles, which are not enclosed within ECVs (Fig. 4). Discussion of the conclusions that can be made with confidence and limitations of the study approaches to distinguish internal from external virus and proteins would be helpful.
The manuscript is generally well-written but clarity may be even further improved with some minor editing.
Author Response
We thank the reviewer for the insightful comments. Responses to the specific comments are in attached file.

Reviewer 2 Report
In this manuscript the authors clearly demonstrate that avian reovirus can be released from infected cells using a nonlytic mechanism, through extracellular vesicles release.
There is accumulating evidence that various non-enveloped viruses that were once (and not so long ago) believed to be strictly released by cell lysis are in fact partly excreted by different cellular exocytosis pathways. The work is, however, original by the virus under study. Avian reoviruses are double-shelled viruses that are similar to mammalian reoviruses, although they do possess a fusogenic protein. Another virus also a member of the Reovirales (rotavirus) was well studied for this aspect as acknowledged by the authors. However, I was a little surprise that there was no mention of the paper by Fernandez de Castro et al on nonlytic egress of mammalian reovirus, although the vesicles involved could be different? This should at least be mentioned and briefly discussed (maybe). There have been reports of extracellular vesicles harboring infectious mammalian reoviruses in meetings (ASV, dsRNA virus meeting) but to my knowledge this has not yet been published, the work presented herein is thus original.
The combination of fractionation, TEM and CryoEM is a strong point of the manuscript.
However, I have a certain number of questions that should be answered in the manuscript and, I believe, should improve its significance.
Specific comments:
•The data are clearly presented and significant in most of the manuscript. However, I noticed the lack of details in description of the methods. This is found at various places and I believed should be corrected, in order to ensure that any competent investigator could replicate the data.
For example, in the ELISA test, which incubation buffer was used, how were the washing steps performed, how long was the incubation with the substrate? Same thing for the western blots (or refer to a previous manuscript in which exactly the same conditions were used). In the qRT-PCR description, it is not clear that the cells were actually infected, and at which MOI?
•In figure 1, overall, the fractionation is well performed and quite convincing. However, I still have a little doubt, is it possible to rule out contamination by residual infected cells, for example by performing a western blot against a cellular protein known to be absent from ECV? Could the authors comment on that? Just a little detail, strictly speaking, the pellet does not contain “virus” but “virions”.
•In figure 2B, it is not clearly explained, why different methods were used and what was the objectives of using different methods at different time points. Nevertheless, the end result is convincing but the manuscript will benefit if it is better explained.
•I am a little confused about the “uncoated virions” that are not cores, how do you explain an abundant presence of these that seems like intermediate in entry but should not be found in large amounts late in the infection. Do the authors believe that they are somehow made by proteolytic enzymes? In the ECVs?
•I do not really understand the objectives of the mass spectrometry experiment, was it necessary to prove the nature of the sigmaC protein?
•The authors present an extensive discussion, with many hypotheses to be tested, in order to explain the higher infectivity of ECV compared to virions, this is quite interesting. However, a trivial explanation could be that the fractionation procedure is somehow detrimental to virion infectivity but not that of the ECV fraction? Is it possible to mention total infectivity before fractionation, is there a loss of total infectivity during the procedure, or not?
Author Response
We thank the reviewer for the insightful comments. Responses to the specific comments are provided in attached file.

Reviewer 3 Report
Nonenveloped Avian Reoviruses Released in Association with Small Extracellular Vesicles are Highly Infectious
Authors describe the role of extracellular vesicles (ECVs) that contained virus particles and their high infectiousness.
The manuscript is relevant and well-written. I have a few minor suggestions below.
Section Materials and Methods.
2.2. “Duncan lab” is better to write the full name. Consider changing the word “purchased”
2.3, 2.4, 2.6. Тhe names in brackets must be complete and the same when it comes to the same company. For example, “GE”- GE Healthcare
2.7. Provide the used protocol, used ingredients and cycling parameters, temperature and etc. If you used reference protocol, then cited it.
Section Results.
Figure 1A, should be separated from others. Иt is scheme, does not show obtained results.
3.1. The last sentence “which suggested that full virus particles or infectious virus core may exist with these ECVs” is part of Discussion.
3.3. “These data imply that the ECVs encompass mostly intact virus particles, and free viral structure proteins, like λA, but not core parti[1]cles”, also is part of Discussion.
Section Discussion.
Second paragraph, third sentence, is not clear and “present” is repeated twice.
Author Response
We thank the reviewer for the constructive comments. Responses to the specific comments are provided in attached file.

Round 2
Reviewer 1 Report
Many viruses, including some viruses in the order Reovirales, are released from cells in association with extracellular vesicles (EVs). Avian reoviruses (ARV) are non-enveloped, fusogenic viruses belonging to the order Reovirales that can be pathogenic in birds, including poultry. Wang et al. used electron microscopy and immunoblotting of ‘EV’ and ‘virus’ fractions released from infected cells, enriched by differential centrifugation and gradient ultracentrifugation, to understand the nature of ARV exit from quail cells (CM5). They report that following ARV infection, CM5 cells release small (~100 nm), electron dense EVs. Their results suggest that ARV proteins associate with EV fractions based on density. To measure the infectivity of ‘EV’ and ‘virus’ fractions, the authors used inocula that were normalized based on outer capsid protein muB or on genome copies to infect cells and quantified viral transcription using RTqPCR or viral progeny by plaque assay. They suggest that association with EVs enhances ARV infectivity up to fifty-fold. Based on the ratios and identities of ARV structural proteins detected in ‘EV’ and ‘virus’ gradient fractions in the presence or absence of detergent treatment and on cryo-EM images of ‘EV’ fractions, they conclude that free ARV proteins are present within EVs, but mature ARV particles associate with but are external to EVs.
Broad comments:
Viral egress mechanisms are understudied, particularly for non-enveloped viruses. Virus egress in association with EVs and its impacts on infection are of broad interest, as they can inform our understanding of virus biology and pathogenesis as well as vaccine design. In the revised manuscript, Wang et al. clarified thoughts about the association of ARV particles (or lack thereof) with small EVs, supplied details that promoted confidence in the approach used to assess ARV protein association with EVs, and substantially improved the discussion. Experiments to test whether the presence of EVs enhances ARV infectivity on a per-genome basis indicate that the presence of EVs can enhance ARV infectivity, measured by the output of infectious ARV by plaque assay. Results obtained through protein normalization and RTqPCR are somewhat less convincing due to high standard deviations but are consistent with a trend towards enhanced ARV infectivity in the presence of EVs. Although the mechanism and relative contribution of EVs to enhancement of ARV infection in a host remain to be determined, this is an intriguing finding.
Specific comments:
2.3 EV isolation: It could be helpful to explain why fractions 14-18 were skipped. It seems reasonable to omit fractions 14-18 to avoid contamination of virions with EVs. However, these fractions appear to contain most of the viral protein, and therefore likely the majority of virus particles released from the cells. The question of whether the presence of a small amount of EVs amidst so many released virus particles is biologically meaningful remains open and might make an interesting contribution to the discussion.
2.8 Infectivity Assay: The authors clarified the ratios of “EV” to “virus” fraction volume used for muB-normalized inocula in the methods by stating that “…the volume of virion fraction used was 2 μL, and the volume of EV fraction ranged from 50-250 μL” If they would please also clarify the ratios of “EV” to “virus” fraction volume used for “virus genome” inocula quantified by RT-qPCR, it would be informative.
Fig. 1D: Do the authors have thoughts about why the amount of sigmaB protein in the pellet by 40 h p.i. is unaffected by detergent treatment, though it is higher at 24 and 28 h in the absence of detergent treatment?
Fig. 1F: “Representative images from one of two experiments are shown.” Please clarify to which panels this statement refers. Based on the figure legend, one might assume it refers only to 1D. However, based on the response to previous reviewer comments, it appears other experiments were conducted multiple times. It could inspire reader confidence to know whether experiments shown in B-D all have been conducted more than once and if the results shown are representative of multiple experiments.
Fig. 2D: Are differences in infectivity statistically significant by a one-sample t-test? Inclusion of this information in the text would be informative. Although not required to support the conclusions as stated, determining whether EVs from uninfected cells alter infectivity when mixed with the ‘virus’ fraction could reveal whether viral proteins contained in EVs or some other property of small EVs is responsible for enhanced infectivity.
This reviewer was unable to access the supplementary video file. The authors might consider adding still tomogram images from the video to the supplement.
Author Response
Dear reviewer,
Please find attached file. Thank you so much for your constructive comments!
Have a good day!
Reviewer 1
Comments and Suggestions for Authors
Many viruses, including some viruses in the order Reovirales, are released from cells in association with extracellular vesicles (EVs). Avian reoviruses (ARV) are non-enveloped, fusogenic viruses belonging to the order Reovirales that can be pathogenic in birds, including poultry. Wang et al. used electron microscopy and immunoblotting of ‘EV’ and ‘virus’ fractions released from infected cells, enriched by differential centrifugation and gradient ultracentrifugation, to understand the nature of ARV exit from quail cells (CM5). They report that following ARV infection, CM5 cells release small (~100 nm), electron dense EVs. Their results suggest that ARV proteins associate with EV fractions based on density. To measure the infectivity of ‘EV’ and ‘virus’ fractions, the authors used inocula that were normalized based on outer capsid protein muB or on genome copies to infect cells and quantified viral transcription using RTqPCR or viral progeny by plaque assay. They suggest that association with EVs enhances ARV infectivity up to fifty-fold. Based on the ratios and identities of ARV structural proteins detected in ‘EV’ and ‘virus’ gradient fractions in the presence or absence of detergent treatment and on cryo-EM images of ‘EV’ fractions, they conclude that free ARV proteins are present within EVs, but mature ARV particles associate with but are external to EVs.
Broad comments:
Viral egress mechanisms are understudied, particularly for non-enveloped viruses. Virus egress in association with EVs and its impacts on infection are of broad interest, as they can inform our understanding of virus biology and pathogenesis as well as vaccine design. In the revised manuscript, Wang et al. clarified thoughts about the association of ARV particles (or lack thereof) with small EVs, supplied details that promoted confidence in the approach used to assess ARV protein association with EVs, and substantially improved the discussion. Experiments to test whether the presence of EVs enhances ARV infectivity on a per-genome basis indicate that the presence of EVs can enhance ARV infectivity, measured by the output of infectious ARV by plaque assay. Results obtained through protein normalization and RTqPCR are somewhat less convincing due to high standard deviations but are consistent with a trend towards enhanced ARV infectivity in the presence of EVs. Although the mechanism and relative contribution of EVs to enhancement of ARV infection in a host remain to be determined, this is an intriguing finding.
Specific comments:
2.3 EV isolation: It could be helpful to explain why fractions 14-18 were skipped. It seems reasonable to omit fractions 14-18 to avoid contamination of virions with EVs. However, these fractions appear to contain most of the viral protein, and therefore likely the majority of virus particles released from the cells. The question of whether the presence of a small amount of EVs amidst so many released virus particles is biologically meaningful remains open and might make an interesting contribution to the discussion.
Response:
We are not sure that we can conclude the majority of virus particles are present in fractions 14-18 even though the majority of the sigmaB outer capsid protein is present in these fractions because the ARV virions are considerably denser than the density of these OptiPrep fractions. We did not investigate the nature of the sigmaB protein present in these fractions.
2.8 Infectivity Assay: The authors clarified the ratios of “EV” to “virus” fraction volume used for muB-normalized inocula in the methods by stating that “…the volume of virion fraction used was 2 μL, and the volume of EV fraction ranged from 50-250 μL” If they would please also clarify the ratios of “EV” to “virus” fraction volume used for “virus genome” inocula quantified by RT-qPCR, it would be informative.
Response:
We added this information in line 190. Generally, 2 uL of the virion fraction gave equivalent genome copies as X-X ul of the EV fraction.
Fig. 1D: Do the authors have thoughts about why the amount of sigmaB protein in the pellet by 40 h p.i. is unaffected by detergent treatment, though it is higher at 24 and 28 h in the absence of detergent treatment?
Response:
We are unsure what the issue is here. Fig 1D does not involve pelleting. The p100 pellet was directly assessed by ELISA, with or without detergent treatment. As we show, the amount of sigmaB protein detected significantly increased in the presence, not the absence, of detergent.
Regarding why the 40 hrs sample does not show increased detection of sigmaB following detergent treatment, we do not know for sure why this is the case, but at this time point most of cells were fused together and starting to lyse (Fig 1B). This is why in all subsequent experiments we indicated that the EVs collection should collected at 24 h post infection.
Fig. 1F: “Representative images from one of two experiments are shown.” Please clarify to which panels this statement refers. Based on the figure legend, one might assume it refers only to 1D. However, based on the response to previous reviewer comments, it appears other experiments were conducted multiple times. It could inspire reader confidence to know whether experiments shown in B-D all have been conducted more than once and if the results shown are representative of multiple experiments.
Response:
We have now indicated in the figure legend that panel1D shows the mean and standard deviation from n=2 independent experiments. Panel 1B is just a time course of syncytium formation in QM5 cells, an experiment we have repeated dozens of times over the years. We have also indicated in the figure legend that Panel 1C was representative of 2 experiments.
Fig. 2D: Are differences in infectivity statistically significant by a one-sample t-test? Inclusion of this information in the text would be informative. Although not required to support the conclusions as stated, determining whether EVs from uninfected cells alter infectivity when mixed with the ‘virus’ fraction could reveal whether viral proteins contained in EVs or some other property of small EVs is responsible for enhanced infectivity.
Response:
We have added statistical significance to Figure 2D.
The suggestion to examine whether EVs from uninfected cells might enhance the infectivity of the virion fraction would be a nice addition to a more detailed future study of the mechanism responsible for the phenotype we have identified on how EVs enhance ARV infectivity.
This reviewer was unable to access the supplementary video file. The authors might consider adding still tomogram images from the video to the supplement.
Response:
We have embedded the tomography video in Supplementary Video 1 to support the still image already included in Figure 3B.

Reviewer 2 Report
Just a little detail, the MRV abbreviation is not defined.
Otherwise this is an extensively revised manuscript that seems as a very useful addition to the literature and quite original, I believe.
Author Response
Dear reviewer,
We thank you so much for your helpful comments!
Just a little detail, the MRV abbreviation is not defined.
Otherwise this is an extensively revised manuscript that seems as a very useful addition to the literature and quite original, I believe.
Response:
Thank you very much for your comments, the abbreviation of MRV has been added.
